# A Mixture of T-Cell Epitope Peptides Derived from Human Respiratory Syncytial Virus F Protein Conferred Protection in DR1-TCR Tg Mice

**DOI:** 10.3390/vaccines12010077

**Published:** 2024-01-11

**Authors:** Hong Guo, Yang Song, Hai Li, Hongqiao Hu, Yuqing Shi, Jie Jiang, Jinyuan Guo, Lei Cao, Naiying Mao, Yan Zhang

**Affiliations:** 1NHC Key Laboratory of Medical Virology and Viral Diseases, National Institute for Viral Disease Control and Prevention, Chinese Center for Disease Control and Prevention, Beijing 102206, China; gh199408@163.com (H.G.); candyalbarn57@126.com (Y.S.); wslihai@126.com (H.L.); huhq049@163.com (H.H.); shiyuqing1997@yeah.net (Y.S.); jiejiang0317@163.com (J.J.); guojy@ivdc.chinacdc.cn (J.G.); caolei@ivdc.chinacdc.cn (L.C.); 2National Key Laboratory of Intelligent Tracking and Forecasting for Infectious Diseases (NITFID), National Institute for Viral Disease Control and Prevention, Chinese Center for Disease Control and Prevention, Beijing 102206, China

**Keywords:** human respiratory syncytial virus, T-cell epitope, peptide vaccine

## Abstract

Human respiratory syncytial virus (HRSV) poses a significant disease burden on global health. To date, two vaccines that primarily induce humoral immunity to prevent HRSV infection have been approved, whereas vaccines that primarily induce T-cell immunity have not yet been well-represented. To address this gap, 25 predicted T-cell epitope peptides derived from the HRSV fusion protein with high human leukocyte antigen (HLA) binding potential were synthesized, and their ability to be recognized by PBMC from previously infected HRSV cases was assessed using an ELISpot assay. Finally, nine T-cell epitope peptides were selected, each of which was recognized by at least 20% of different donors’ PBMC as potential vaccine candidates to prevent HRSV infection. The protective efficacy of F-9PV, a combination of nine peptides along with CpG-ODN and aluminum phosphate (Al) adjuvants, was validated in both HLA-humanized mice (DR1-TCR transgenic mice, Tg mice) and wild-type (WT) mice. The results show that F-9PV significantly enhanced protection against viral challenge as evidenced by reductions in viral load and pathological lesions in mice lungs. In addition, F-9PV elicits robust Th1-biased response, thereby mitigating the potential safety risk of Th2-induced respiratory disease during HRSV infection. Compared to WT mice, the F-9PV mice exhibited superior protection and immunogenicity in Tg mice, underscoring the specificity for human HLA. Overall, our results demonstrate that T-cell epitope peptides provide protection against HRSV infection in animal models even in the absence of neutralizing antibodies, indicating the feasibility of developing an HRSV T-cell epitope peptide-based vaccine.

## 1. Introduction

Human respiratory syncytial virus (HRSV) is the predominant pathogen causing acute lower respiratory infections (ALRIs) in infants and children [1]. In 2019, there were 33 million ALRIs associated with HRSV worldwide, 3.6 million hospital admissions, and 26,300 fatalities aged 0–60 months [2]. HRSV belongs to the *Orthopenumovirus* genus in the family of *Pneumoviridae* and can be divided into two subtypes, HRSV-A and HRSV-B [3]. Its genome consists of negative-sense, single-strand RNA with a length of approximately 15,200 nucleotides and it encodes at least 11 proteins [4,5]. Of these, two major glycoproteins, attachment glycoprotein (G) and fusion (F) protein, are crucial for viral infectivity and pathogenesis [6]. In particular, the F protein is a preferred target for vaccine research due to its important role in the viral invasion of host cells and its relative conservation in the two subtypes of HRSV [7]. As of August 2023, two F protein subunit vaccines for the prevention of HRSV infection, Arexvy (RSVPreF1, GSK) and Abrysvo (RSVpreF, Pfizer), have been approved by the US Food and Drug Administration (FDA) [8]. Both vaccines protect against HRSV infection primarily by inducing humoral immunity that produces effective neutralizing antibodies [9,10]. However, HRSV-specific antibodies wanes over time, and even individuals with the highest antibody levels have a 25% risk of reinfection. A vaccine that relies solely on neutralizing antibodies as the primary correlate of protection against HRSV infection may not provide universal or durable immunity [11,12].

Human T-cells are crucial in managing HRSV infection, clearing intracellular pathogens, and maintaining long-lasting immune responses [13,14,15]. Moreover, primed human CD8^+^ T-cells or CD4^+^ T-cells effectively and independently control HRSV replication in human lung tissue in the absence of an HRSV-specific antibody response [16]. Consequently, an optimal vaccine can induce both humoral and cellular immunity, thereby eliciting a balanced immune response in the host. However, the current emphasis of most vaccines remains predominantly on humoral immunity, with the absence of comprehensive research directed towards cellular immunity [17]. The F protein, situated on the virion surface, is unique as it is the only highly conserved surface antigen across all known HRSV subtypes that counteract immune evasion caused by mutations in HRSV [18,19]. The HRSV F protein contains cytotoxic T lymphocyte (CTL) epitopes recognizable in both humans and mice [20]. Several studies have identified important epitopes on the F protein that induce the secretion of T helper 1 (Th1) cytokines, specifically IL-2, IL-12, and IFN-γ, which are key targets for human memory CD4^+^ T-cells [14,21,22]. Thus, targeting the cell immune epitopes of F protein might be an effective approach to developing vaccine candidates against HRSV infection.

Peptide-based vaccines are synthesized in vitro by utilizing known immunogenic amino acid sequences, generally from B-cell or T-cell epitopes, and they have recently gained attention due to their ability to trigger both humoral and T-cell immune responses [23,24,25]. They have been used for the development of vaccines against various infectious diseases, including Malaria, influenza, HIV-1, SARS-CoV-2, and hepatitis C virus [23,26,27,28]. Vaccines based on T-cell epitope peptides focus T-cell responses on a specific set of critical epitopes, and the selection of minimal epitopes or short domains of the antigen of interest avoids the deleterious functions that are sometimes encountered when whole antigens are used [25]. The identification of immunogenic epitopes among the potential epitopes present in an antigenic protein is thus a critical step in attempts to optimize T-cell mediated immunity and is important for the development of efficient vaccines. However, the identification of specific epitopes from complex antigens can be a cumbersome and difficult process due to the multiplicity of the major histocompatibility (MHC) molecules encoded by distinct gene loci in both mice and humans [29]. Despite identifying numerous T-cell epitopes on the HRSV F protein, there is a shortage of relevant animal experiments to confirm their effectiveness in providing protection. Although humanized mice cannot fully replicate the complexity and dynamics of the human immune system, they can simulate certain aspects of the human immune response. Humanized mice have been adopted as a promising model to study human immunity [30]. The DR1-TCR Tg mice used in this study contain the HLA-DR1 gene and can be used as antigen-specific T-cells for immunological studies.

In this study, a total of nine T-cell epitope peptides, which contain conserved amino acid (AA) sequences of HRSV F protein and cover most of the HLA genotypes found in the Chinese population were screened and carefully evaluated using an animal model. Viral challenge assays showed that the nine-peptide mixture effectively suppressed viral replication in the lungs and notably alleviated lung injury. Importantly, our findings revealed that the T-cell epitope peptide mixture can confer robust challenge protection in HLA humanized mice without inducing an enhanced disease response. These results provide a promising foundation for the development of a T-cell peptide-based vaccine against HRSV infection.

## 2. Materials and Methods

### 2.1. PBMC, Mice, and Ethics

Peripheral blood mononuclear cells (PBMCs) were obtained from 28 donors in our laboratory who had HRSV infections and had significantly elevated serum levels of HRSV IgG antibody (≥1.55 U/mL), with their informed consent. All procedures were conducted in accordance with the tenets of the Declaration of Helsinki. C57BL/6J mice that were specific-pathogen-free (SPF) and male were procured from Beijing Vital River Laboratory Animal Technology Co., Ltd. (Beijing, China), DR1-TCR Tg mice (C57BL/6J background) were obtained from The Jackson Laboratory (stock: 026566), with a detailed generation description previously reported [31]. The mice used in these studies were maintained in a pathogen-free environment and fed standard rodent chow (Ralston Purina) and water ad libitum in the Animal Experimental Center of the Chinese Center for Disease Control and Prevention (China CDC). All animals were maintained until they reached the age of 7–10 weeks prior to the experiments. This study had been approved by the Ethics Committees: the National Institute for Viral Disease Control and Prevention, China CDC. (No. 20220408040; 20220408040-BG).

### 2.2. Peptide Screening and Synthesis

Peptides were initially selected from the IEDB public database as of December 2021. To retrieve T-cell epitopes, the epitope source was restricted to “Human Respiratory syncytial virus, Human respiratory syncytial virus A (Respiratory syncytial virus group A), Human respiratory syncytial virus B (Respiratory syncytial virus group B)”, and “Human” was selected from the “Host” section and “T-cell” from the “Assay” section.

According to the above conditions, after initially collecting the epitopes, further selection was made based on the published literature and the Allele Frequency Net Database to synthesize T-cell epitopes located on the F protein, which have a relatively conserved AA and high potential for HLA binding. The selected F protein T-cell peptides were synthesized by Nanjing GenScript Biotechnology Co., Ltd. (Nanjing, China) with a purity of 95% and analyzed using high performance liquid chromatography (HPLC) to ensure their quality. The peptides were diluted in sterile water to a storage concentration of 1 mg/mL for use in subsequent experiments.

The ELISpot assay was applied to detect T-cell responses against HRSV F protein peptides using ELISpot Plus: Human IFN-γ kit (3420-4HPW, Mabtech, Nacka Strand, Sweden). Plates were washed three times with PBS and then conditioned with serum-free medium (6015012, Dakewe Bioengineering Co., Ltd., Beijing, China) for ELISpot for 30 min at room temperature (RT). We added 2.5 × 10^5^ cells/well PBMCs (in 100 µL) from different samples to wells. PBMCs were stimulated in vitro with selected peptides (10 µg/mL for each peptide, 10 µL/well) for 36 h in an incubator at 37 °C containing 5% CO_2_. Serum-free medium was used as a negative control, and phytohemagglutinin (PHA) as a positive control. Detection proceeded according to the manufacturer’s instructions. After drying, the IRIS Mabtech ELISpot/FluoroSpot reader was used to count spot-forming cells (SFCs). Subsequent experimental procedures were performed following the manufacturer’s instructions. Finally, we stored the plate in the dark at room temperature until it dried completely. The SFCs were counted under an IRIS Mabtech ELISpot/FluoroSpot reader. Positive IFNγ-ELISpot response was defined as at least five spot-forming cells per 1 × 10^5^ PBMCs and a minimum of a two-times increase from baseline in each sample. Furthermore, 28 donors’ PBMCs were individually stimulated with each peptide, and peptides that could elicit a positive response in ≥20% of the donors were selected. We mixed the positive peptides in equal proportions (0.3 µg/mL). All peptides were stored at −80 °C until utilized.

### 2.3. Immunization and HRSV-A Long Challenge

Male DR1-TCR Tg mice, aged 7–10 weeks, were divided into two groups: the mixed peptide immunized group (DR1-F9) and adjuvant control group (DR1-ADJU). Similarly, male C57BL/6J mice of the same age range were divided into the mixed peptide immunized group WT-F9 group and the adjuvant control WT-ADJU group. Animal classifications and vaccine details can be found in Figure 1B. The same strain of mice were randomly sorted into groups of four and immunized subcutaneously at the base of the tail on days 0 and 21. Mice were immunized with a total volume of 100 μL containing 270 μg of mixed F protein peptide, 40 μg of 59 cytosine-phosphate-guanine 39 oligodeoxynucleotide (CpG-ODN; referred to here as CpG) (Parr Biotechnology Jiangsu Co., Ltd., Nanjing, China), and 100 μg of Adju-Phos adjuvant (Croda). Conversely, the control groups received two subcutaneous injections consisting of 270μg phosphate-buffered saline (PBS), 40 μg CPG, and 100 μg Adju-Phos adjuvant. On the 35th day (14 days after the second dose), all mice were intranasally challenged with 50 μL of HRSV-A Long, equivalent to 1 × 10^6^ plaque-forming units (PFU), under the sedative influence of isoflurane anesthesia. Mice were monitored daily for symptoms and body weight changes. On day 4 post-infection (dpi), the subjects were euthanized, and their blood, lung, and spleen tissues were collected to evaluate immunological responses (Figure 1).

### 2.4. Virus Neutralization

One day before the neutralization assay, in each well of 96-well plates, we added 100 μL of HEp-2 cells (ATCC, CCL-23) at a density of 2 × 10^5^ cells/mL. The following day, sera from immunized mice were inactivated at 56 °C for 30 min. Then, the sera were diluted 2-fold in cell culture medium, starting at 1:5 using 96-well plates. The diluted sera (100 mL/well) were mixed with an equal volume of diluted HRSV (10^4^ PFU) and incubated at 37 °C with 5% CO_2_ for 30 min to neutralize the virus. The mixture of virus and serum was transferred to the 96-well plates containing HEp-2 monolayers. The cells were fixed using 4% Paraformaldehyde Fix Solution (DF0133, Beijing Leagene Biotech. Co., Ltd., Beijing, China) for 15–20 min after 30 h post-infection. The plates were washed and blocked with PBS-1% bovine serum albumin (BSA) and 0.1% Tween 20 for 30 min at 37 °C. After washing again, the fixed cells were incubated first with 1:1000-diluted anti-F monoclonal antibodies (palivizumab, expressed and purified by our laboratory) in PBS (100 μL/well) and then incubated with 1:2000-diluted horseradish peroxidase-labelled Goat anti-Human IgGoat anti-Human IgG (H + L) (ZSGB-BIO, ZB2304). Both the primary and secondary antibody incubations each required 1 h at 37 °C. The cells were washed with PBST and stained with TrueBlue Peroxidase Substrate. The neutralization titer was determined using Karber’s method, which calculated the logarithm of the maximum dilution required to reduce virus infectivity by 50%.

### 2.5. FluoroSpot in Splenic Lymphocytes

FluoroSpot assays were used to profile the cytokines secreted by T-cells, providing insight into the Th1 or Th2 bias of splenic lymphocytes. On day 35, the mice were euthanized and their spleens were aseptically collected. Splenocytes were then isolated using Mouse 1× Lymphocyte Separation Medium (7211011, Dakewe Bioengineering Co., Ltd., Beijing, China). The FluoroSpot Plus kits (FSP-4146, FSP-414245, and P-414743; Mabtech, Sweden) were distributed at a density of 5 × 10^5^ cells/well to detect different cytokines. Th1 responses include IFN-γ, IL-2, and TNF-α, while Th2 responses are scrutinized by IL-4 and IL-10; all cytokines were determined in parallel experiments. Controls and stimulations mirrored the ELISpot procedure, with splenocytes exposed to the F protein peptide mixture (10 mg/mL/peptide) for 36–40 h at 37 °C in a 5% CO_2_ humidified incubator. The subsequent steps conformed to manufacturer’s guidelines. The dried plates were stored shielded from light at RT and SFCs were tallied using the IRIS Mabtech ELISpot/FluoroSpot reader.

### 2.6. Viral Load in Lung Tissue

Each right lung was aseptically removed and homogenized in 0.5 mL of PBS containing 30% sucrose. After centrifuging the cellular debris at 1500 rpm for 10 min, 200 μL of the supernatant was used for nucleic acid extraction using a commercial Kit (Xi’an Tianlong, Xi’an, China). The viral load of HRSV in the lungs of mice was quantified by digital polymerase chain reaction (dPCR) [32]. The Naica System™ for Crystal Digital™ PCR (Stilla Technologies, Villejuif, France) was used for analysis.

### 2.7. Histopathological Analysis in Lung Tissue

Each left lung of the mice was fixed using 4% formalin and then embedded in paraffin. Sections were prepared and stained with hematoxylin and eosin (H&E). According to the morphological changes following HRSV challenge, the lung tissue was graded into four categories, including normal (0), mild (1), moderate (2), severe (3), and life-threatening (4). A blinded histopathology expert scored the lung tissue based on the inflammatory cell infiltration, parenchymal pneumonia, alveolar hemorrhage, and bronchiolar/bronchial luminal or alveolar exudate.

### 2.8. Statistical Analyses 

The data were analyzed using GraphPad Prism 9.0 software (GraphPad Software, Boston, MA, USA). The statistical significance of the virological and immunological outcomes was determined using either one-way or two-way ANOVA, followed by Fisher’s LSD test. Values of *p* < 0.05 were considered significant; individual *p* value comparisons between groups are reported in figure legends.

## 3. Results

### 3.1. Nine T-Cell Epitopes Were Selected from HRSV-F Protein

A total of 119 epitopes were initially gathered from the IEDB public database (Appendix A). By reviewing experimental data from publications and selecting T-cell epitopes located on F proteins with relatively conserved AA and high peptide-HLA binding potential, 25 T-cell epitopes were screened and synthesized. 

Out of the 25 peptides, 9 T-cell epitopes were ultimately selected based on their positive response rate of ≥20% among the 28 donors. These nine peptides were F-6 (10, 35.71%), F-12 (7, 25%), F-13 (8, 28.57%), F-16 (8, 28.57%), F-17 (6, 21.43%), F-19 (11, 39.29%), F-22 (11, 39.29%), F-23 (11, 39.29%), and F-24 (14, 50%). The nine-peptide mixture from the F protein was named ‘F-9P’ (Figure 1A). Subsequently, PBMCs from 28 donors were separately stimulated with F-9P. The results demonstrated that a significant immune response was elicited in 64.29% (18/28) of the donors (Table 1). Building upon these findings, F-9P was further formulated by combining it with CpG and AL adjuvants to create the ‘F-9P Vaccine (F-9PV)’ for subsequent immunization studies conducted on animals. 

### 3.2. F-9PV Induced a Th-1-Biased T-Cell Response in HLA-DR1 Tg Mice

Mice were immunized subcutaneously with F-9PV. After adhering to the challenge protocol (Figure 1B), splenocytes that were extracted from mice 4 days following HRSV challenge were analyzed for Th1- and Th2-baised responses to peptide-specific cellular immunity (Figure 2).

In DR1-TCR Tg mice, the DR1-F9 group showed significantly higher levels of the cytokines including IFN-γ (95.00% CI, 211.6 to 285.4; *p* < 0.0001), IL-2 (95.00% CI, 21.95 to 64.05; *p* = 0.0008), TNF-α (95.00% CI, 93.16 to 200.8; *p* < 0.0001), IL-4 (95.00% CI, 23.29 to 76.71; *p* = 0.0015), and IL-10 (95.00% CI, 35.41 to 101.6; *p* = 0.0007) compared to the DR1-ADJU group (Figure 2A–E). Similarly, in WT mice, the WT-F9 group also demonstrated higher levels of cytokines IFN-γ (95.00% CI, 49.34 to 123.2; *p* = 0.0003), TNF-α (95.00% CI, 3.659 to 111.3; *p* = 0.0383), and IL-10 (95.00% CI, 9.412 to 75.59; *p* = 0.0161) compared to the WT-ADJU group (Figure 2A,B,D). These results suggest that F-9P can effectively stimulate T-cell immune response in mice. 

We compared the differences between F-9PV-immunized DR1-TCR Tg and WT mice. The DR1-F9 group produced a greater quantity of IFN-γ (95.00% CI, 114.6 to 188.4; *p* < 0.0001), IL-2 (95.00% CI, 5.446 to 47.55; *p* = 0.0179), and TNF-α (95.00% CI, 32.91 to 140.6; *p* = 0.0043) than the WT-F9 group (Figure 2A–C). This indicates that F-9P is more targeted towards human MHC-II (HLA-II) rather than mouse MHC-II (H-2).

Additionally, the DR1-F9 group displayed the highest ration of IFN-γ to IL-4 secretion among the four groups (Figure 2F), indicating that F-9PV induced a significantly Th1-biased cellular immune response in DR1-TCR Tg mice. The levels of neutralizing antibody against the HRSV-A Long strain were quantified in mouse serum at 4 dpi, and all groups exhibited neutralizing antibody titers (<1:10) lower than the initial dilution.

### 3.3. F-9PV Provided Protective Efficacy against HRSV Infection

F-9PV vaccination demonstrated protective efficacy against HRSV infection, as evidenced by daily monitoring of mice for symptoms and body weight changes post-infection. On the first day after challenge, all groups of mice experienced a decrease in body weight, followed by gradual recovery at 2 dpi. By the fourth day, only the DR1-F9 group of mice fully restored their body weights to their initial levels. Furthermore, when viral load of the lung was assessed on 4 dpi, the HRSV RNA copies of the DR1-F9 group and WT-F9 group were lower than those in the adjuvanted control group (*p* < 0.05), and the DR1-F9 group exhibited the lowest level of HRSV RNA copies (Figure 3B).

Pathology scores were also evaluated, with both F-9PV-immunized groups exhibiting lower pathology scores compared to the adjuvanted groups (Figure 3C). The DR1-F9 group vaccinated with F-9PV demonstrated mild alveolar wall thickening and epithelial hyperplasia, while the DR1-ADJU and WT-ADJU group mice exhibited significant lung pathology, including widened alveolar septa, septal fusion, narrowed alveolar spaces, and localized lung tissue consolidation (Figure 3D). The lung damage was notably severe in the adjuvanted groups, with two out of the DR1-ADJU group and three out of the WT-ADJU group mice exhibiting peripheral inflammatory cells dominated by lymphocytes and fibrinous exudate, along with some bronchiolar epithelial cells shedding into the lumen (Figure 3C). There was a correlation between pathological scores and viral load. The DR1-F9 group showed mild morphological changes with a total pathology score of 1 and had the lowest viral load in lung tissue of all groups at 10^1.2^ copies/μL. In contrast, the WT-ADJU group had the highest lung pathology score of 2.7, with a degree of pathology between moderate and severe. Additionally, the WT-ADJU group also had the highest viral load of the four groups at 10^2.1^ copies/μL. Overall, F-9PV vaccination effectively reduced viral load and minimized pathological lung damage, suggesting that F-9PV might provide protection against HRSV infection in mice by inducing effective cellular immunity, delaying disease progression, and attenuating lung inflammation.

## 4. Discussion

The MHC molecules (known as HLA in humans) that restrict antigen epitopes recognized by T-cells are critical to the development of peptide-based vaccines [26]. There are two main classes of MHC molecules: MHC I expressed on the surfaces of all nucleated cells and MHC II found on the surfaces of specialized APCs. Two types of T-cells are specially equipped for binding to the MHC I and II, the CD8^+^ and CD4^+^ T-cells, respectively [34]. The T-cell epitope recognition is variable within the population due to the high level of polymorphism in the genes encoding HLA [35]. Previous studies have shown that the selection of MHC II-restricted antigen targets recognized by T-cells is critical for clinical outcomes in preventing HRSV infection [36]. In our study, 25 linear epitope peptides, consisting of 9 to 18 amino acids with high binding affinity to HLA molecules, were synthetized. These predicted peptides with high peptide-HLA allele binding affinity, located at different sites on the F protein, had different HLA-II restrictions, such as DRB1*15:01, DRB1*07:01, HLA-DRB1*15:02, DRB1*01:01, and DRB1*04:01 (Appendix A). According to the Allele Frequency Net Database, the common HLA alleles in Asia include HLA-DRB1*01:01, HLA-DRB1*15:01, and HLA-DRB1*15:02, while common HLA alleles in Europe are HLA-DRB1*07:01, HLA-DRB1*15:01, and HLA-DRB1*04:01. While prediction algorithms can effectively narrow down the pool of potential epitopes for further investigation, experimental validation is crucial to confirm their actual immunogenicity and functionality [37]. Among the 25 predicted T-cell epitope peptides, only 9 peptides were recognized by more than 20% of PBMC cells from previously infected HRSV donors; presumably these epitopes could be recognized by the memory T-cells of infected donors and can induce an effective T-cell immune response. To overcome the limited genotype of HLA corresponding to a single peptide, nine T-cell peptides were mixed to increase the coverage of the HLA allelic populations, resulting in a positive reaction rate to donor PBMC of 64.29%.

Given the significant T-cell MHC differences between humans and mice, DR1-TCR transgenic (Tg) mice carrying the HLA-DR1 transgene were investigated. Our results showed that the T-cell immune response induced by F-9PV significantly reduced viral load and pathological changes in mouse lung tissue, and provided protection against HRSV challenge, even in the absence of neutralizing antibodies. Compared to the adjuvant only control group (DR1-ADJU), F-9PV-immunized mice produced higher levels of cytokines. In particular, the secretion of IFN-γ in the DR1-F9 group was 17 times higher than in the DR1-ADJU group, and the ratio of IFN-γ/IL-4 was 6.4 times higher than in the DR1-ADJU group. These results showed that F-9PV induces a significant Th1-biased T-cell immune response in mice, indicating the potential immunoregulatory function of F-9PV. Previous studies have shown that a Th1-biased T-cell immune response produces cytokines such as IFN-γ at the site of infection, thereby reducing the replication of HRSV in airway epithelial cells [38]. In contrast, the Th2-biased immune response was observed to be associated with vaccine-associated enhanced respiratory disease (ERD) caused by formalin-inactivated RSV (FI-RSV) vaccines in clinical trials [39]. Therefore, T-cell peptides combined with an adjuvant may induce a protective immune response against HRSV infection in murine mode and prevent the ERD triggered by an unbalanced Th2-biased immune response upon subsequent exposure to HRSV. However, the roles of the individual peptides involved the T-cell immune response need to be further investigated. In addition, compared to WT mice, Tg mice immunized with F-9PV induced a stronger T-cell immune response, suggesting that F-9PV might have a high binding affinity to human HLA genotypes. However, WT mice also showed a mild T-cell immune response, which might be related to the T-cell epitopes shared by humans and mice.

To date, at least 45 genotypes have been identified worldwide, demonstrating the genetic diversity of HRSV [40,41,42,43]. An effective broad-spectrum vaccine targeting the conserved amino acid region is therefore required to avoid potential immune escape through mutations. Among the 11 encoded proteins of the HRSV genome, the F protein is highly conserved with nearly 90% homology between HRSV-A and HRSV-B [44]. In our study, T-cell epitope peptides were designed on conserved regions of the F protein, in which all nine peptides are completely conserved in HRSV-A, and three of them are also fully conserved in HRSV-B. The remaining six peptides shared at least 89% homology with HRSV-B. The F protein contains three functional regions, which can induce humoral and cellular immune responses after HRSV infection [45]. F-9PV contained two peptides located in the F2 region, one in the P27 region and six in the F1 region. Of these, one peptide F13 (AA 391–405) was located in the neutralizing site I (AA 380–400), F22 (AA 55–72) in the neutralizing site Ø (AA 62–69 and AA 196–209), and F23 (AA 409–426) in the neutralizing site V (AA 422–438). F-9PV did not induce notable levels of neutralizing antibodies in mice, possibly due to the limited coverage of the neutralizing site and the liner structure of these three peptides. However, our results showed that the F-9PV vaccine, which induces with a Th1-biased response, had fewer lung pulmonary pathological changes after viral challenge, suggesting that only T-cell responses can induce protective immunity in mice. 

This study has some limitations. (i) All the peptides used in our study belong to linear epitopes, which have a limited ability to induce protective humoral immune responses. Therefore, to induce stronger and more diverse immune responses, F-9PV might need to be combined with other humoral immune-inducing peptides or protein subunits. (ii) Due to the mixture of peptides used in our study, the immune response capacity of each peptide in F-9PV could not be determined. (iii) The HRSV-A Long strain was used as a challenge virus without comparing it to the protective effect of subtype B. Whether peptide induced cellular immunity also protects against different viral genotypes needs to be further investigated [46]. (iv) Transgenic mice were used to mimic the human immune response in our study. It is the unclear to what extent the immune response in humanized mice accurately reflects the human immune response. Further research is needed to determine whether the T-cell peptide vaccine developed in this study can also induce effective cellular immunity in humans.

A previous study showed that the CoVac-1 vaccine based on the SARS-CoV-2 T-cell epitope could induce strong and diverse T-cell responses in patients with B-cell or antibody deficiencies, suggesting that T-cell immunity alone could control the viral infections [47]. To enhance vaccine efficacy, comprehensive stimulation of both humoral and cellular responses by a combination of T-cell epitope peptide-based antigens and traditional neutralizing antibody-based vaccines was conducted, which demonstrated efficacy and feasibility in SARS-CoV-2 vaccines [48,49,50]. In this study, we demonstrated that a peptide-based mixture, F-9PV, derived from conserved T-cell epitopes of the F protein of HRSV, could prevent HRSV infection in DR1-TCR transgenic mice by inducing a Th1-biased T-cell immune response, indicating the feasibility of developing an HRSV T-cell epitope peptide-based vaccine. 

## 5. Conclusions

In conclusion, our findings from this study indicate that the T-cell epitope peptide mixture can confer robust protection in DR1-TCR Tg mice, effectively suppress viral replication in the lungs, and notably alleviate lung injury from the challenge. Therefore, we highlight a promising approach using T-cell hybrid peptides to generate effective cellular immunity as a strategy to optimize HRSV vaccine design and improve vaccine efficacy.

## Figures and Tables

**Figure 1 vaccines-12-00077-f001:**
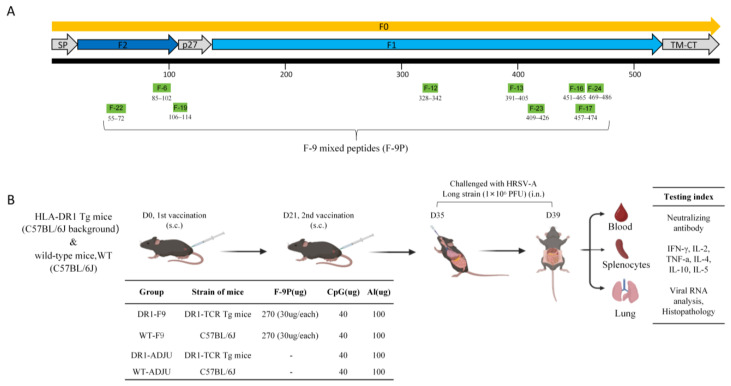
Schematic representation of groups, immunization procedure, and challenge schedule. (**A**) Linear diagram of the HRSV fusion (F) protein ectodomain based on the protein sequence of HRSV-A Long. Site of the nine peptides with more than 20% positivity on the F protein. The nine-peptide mixture from the F protein was named ‘F-9P’ (**B**) DR1-TCR Tg mice and C57BL/6J were subcutaneously injected with either F-9PV (*n* = 4) or adjuvant buffer (*n* = 4) on days 0 and 21. On day 35, they were challenged with 1 × 10^6^ PFU of HRSV-A Long. Mice were monitored daily for symptoms and body weight changes. On day 4 post-infection, blood, lung, and spleen tissues were harvested for neutralizing antibody, histological assessments, viral load analyses, and immune response assays.

**Figure 2 vaccines-12-00077-f002:**
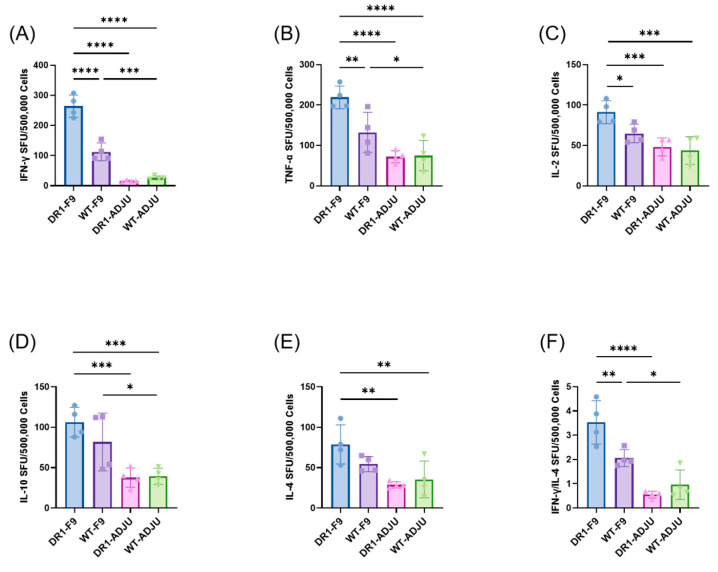
Mice exhibited Th1-biased T-cell responses. Mice splenocytes harvested 4 days after HRSV challenge were stimulated with F-9 PV for (**A**) IFN-γ spot-forming unit (SFU), (**B**) TNF-α SFU, (**C**) IL-2 SFU, (**D**) IL-10 SFU, (**E**) IL-4 SFU, and (**F**) IFN-γ/IL-4 SFU, respectively. Points represent individual mice. Statistically significant differences were measured by one-way ANOVA with Fisher’s LSD test. **** *p* < 0.0001, *** *p* < 0.001, ** *p* < 0.01, * *p* < 0.05.

**Figure 3 vaccines-12-00077-f003:**
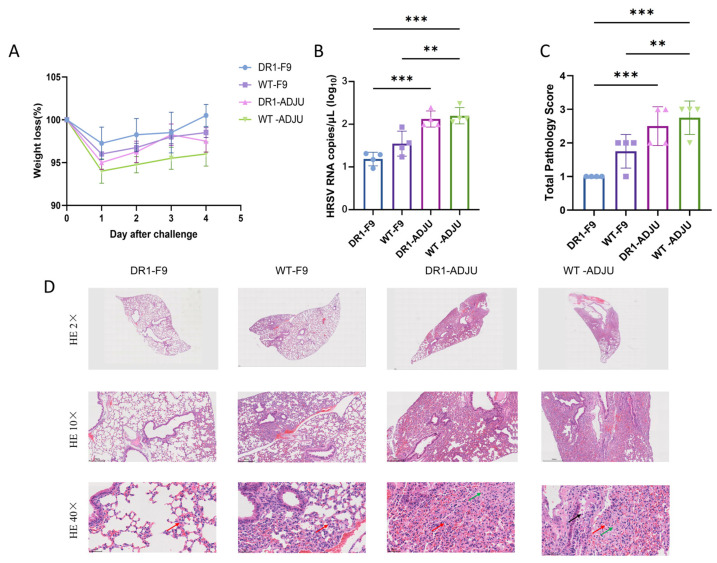
F-9PV provided protective efficacy against HRSV infection. Mice were monitored daily for symptoms and body weight changes. On day 4 dpi, the subjects were euthanized, and their lung tissues were collected to evaluate immunological responses. Points represent individual mice. (**A**) The same group of immune mice were challenged with HRSV and then weighed each day. (**B**) The copies of HRSV RNA in the lung. (**C**) Scoring of total pathology after HRSV challenge of immunized mice. (**D**) Left lungs histopathological analysis from adjuvant control and peptide mixture-treated DR1-TCR Tg mice and C57BL/6J at 4 dpc. Red arrows indicate inflammatory cell infiltrate dominated by lymphocytes. Green arrows indicate fibrinoid exudation. Black arrows indicate some bronchiolar epithelial cells shed into the lumen. H&E staining. Bars: 2×, 800 μm to 1000 mm; 10×, 200 μm; 40×, 60 μm. Results are expressed as the mean ± SEM from 4 mice for each group. *** *p* < 0.001, ** *p* < 0.01.

**Table 1 vaccines-12-00077-t001:** IFN-γ production in PBMCs from donors induced by peptide frequency.

Peptide	IEDB ID	Peptide Sequence	Site ^#^	Positive Subjects (*n* = 28)	Positive Rate (%)	Reference
F-6	153633	KYKNAVTELQLLMQSTPP	85–102	10	35.71% (10/28)	[22]
F-12	99202	EGSNICLTRTDRGWY	328–342	7	25.00% (7/28)	[33]
F-13	99780	YDCKIMTSKTDVSSS	391–405	8	28.57% (8/28)	[33]
F-16	99691	SVGNTLYYVNKQEGK	451–465	8	28.57% (8/28)	[33]
F-17	153747	YYVNKQEGKSLYVKGEPI	457–474	6	21.43% (6/28)	[22]
F-19	53201	RARRELPRF	106–114	11	39.29% (11/28)	[21]
F-22	153719	SVITIELSNIKENKCNGT	55–72	11	39.29% (11/28)	[22]
F-23	153709	SLGAIVSCYGKTKCTASN	409–426	11	39.29% (11/28)	[22]
F-24	153737	VKGEPIINFYDPLVFPSD	469–486	14	50.00% (14/28)	[22]
F-9P	NA	NA	NA	18	64.29% (18/28)	

^#^ The site located in F protein based on the sequence of HRSV-A Long. NA, nine-peptide-based mixture of F-9PV not yet analyzed by the IEDB.

## Data Availability

Data are available upon request by mailing the first author.

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
