# Peer review of "A Mixture of T-Cell Epitope Peptides Derived from Human Respiratory Syncytial Virus F Protein Conferred Protection in DR1-TCR Tg Mice"

_vaccines, 2024, doi:10.3390/vaccines12010077_

Round 1

Reviewer 1 Report

Comments and Suggestions for Authors

The manuscript “A Mixture of T Cell Epitope Peptides Derived from HRSV F Protein Conferred Protection in DR1-TCR Tg Mice” addresses a significant gap in the field of respiratory syncytial virus (HRSV) vaccine development by focusing on T-cell epitope-based immunization strategies. T-cell immunity plays a crucial role in combating viral infections, and the study's emphasis on identifying conserved T-cell epitopes within the highly conserved F protein of HRSV is commendable. The use of transgenic mice expressing human leukocyte antigen (HLA) molecules enhances the translational relevance of the findings. The demonstrated protective efficacy against HRSV infection, as evidenced by reduced viral loads and lung pathology in immunized mice, highlights the potential of T-cell epitope-based vaccines. Moreover, the manuscript suggests a Th1-biased immune response induced by the formulated vaccine, which is crucial for effective antiviral immunity. The identification of specific T-cell epitopes and their formulation into a vaccine offers a promising avenue for developing a broadly protective and potentially universal HRSV vaccine.

1.     The manuscript generally maintains a good standard of grammatical accuracy, but a few noteworthy errors require attention. There are instances of inconsistent verb tense usage, particularly in the presentation of results. For example, in the Results section, past tense is used to describe the synthesis and screening process of T-cell epitopes, but then present tense is employed when discussing the subsequent formulation and immunization studies. Maintaining a consistent tense throughout would enhance the overall clarity and coherence of the narrative.

2.     While the abstract addresses the critical need for T-cell-mediated immunity against Human Respiratory Syncytial Virus (HRSV) and presents a comprehensive approach to identify potential vaccine candidates, several limitations should be considered.

3.     The introduction provides a comprehensive overview of the significance of Human Respiratory Syncytial Virus (HRSV) and the current limitations of existing vaccines, paving the way for the exploration of T-cell epitope-based alternatives. However, several shortcomings merit attention. The introduction lacks a clear statement on the novelty of the research, and while it underscores the importance of T-cell responses, it could better emphasize the need for a more balanced immune response, integrating both humoral and cell-mediated immunity.

4.     The study's reliance on predicted T-cell epitopes raises concerns about the accuracy of these predictions, and a more detailed discussion on the potential challenges associated with in silico epitope identification is warranted. Furthermore, the introduction would benefit from a more explicit connection between the identified T-cell epitopes and their ability to induce a protective immune response, addressing the critical question of how these epitopes contribute to effective immunity against HRSV.

5.     Additionally, the introduction could provide more context on the limitations and advantages of using humanized mice as an animal model, acknowledging its relevance but also its inherent differences from the human immune system.

6.     The materials and methods section provides a detailed account of the experimental procedures, but several shortcomings need consideration. Firstly, the source of peripheral blood mononuclear cells (PBMCs) from 28 donors with HRSV infections is mentioned, but there is a lack of information regarding the selection criteria for these donors, potentially introducing variability in the immune responses observed.

7.     The results section presents comprehensive data on the selection of T-cell epitopes, the immune response induced by F-9PV vaccination, and the protective efficacy against HRSV infection. However, the interpretation of the pathology scores and histopathological analysis lacks a more in-depth discussion, and a quantitative analysis of the lung damage would strengthen the conclusions drawn. Addressing these points would enhance the clarity and rigor of the study.

8.     The discussion provides a detailed analysis of the T-cell epitope selection, immune responses induced by F-9PV vaccination, and the potential implications for developing an HRSV vaccine. However, the study lacks a more thorough exploration of the limitations associated with the use of linear epitopes and potential challenges in translating findings from mice to humans.

9.     The discussion on the immune response in mice primarily focuses on cytokine production, and more information on other aspects of the immune response, such as memory T-cell generation and the absence of neutralizing antibodies, would enhance the depth of the analysis.

10.  Additionally, the study acknowledges the genetic diversity of HRSV but does not extensively discuss the implications of this diversity for the efficacy of the vaccine across different viral genotypes.

Comments on the Quality of English Language

Minor Editing is required.

Reviewer 2 Report

Comments and Suggestions for Authors

The manuscript is devoted to the topical theme of designing a specific and effective vaccine against HRSV and contains original data obtained by the authors. Design of T-epitope-based peptide vaccines directed to the stimulation of cellular immuno responses is an obvious way for protecting against viral infections.

1. However, since F protein-based vaccines eliciting specific antibody, i.e. humoral, responses against HRSV are already under use, a more profound explanation of the desirability of the design of T-cell epitope peptide-based vaccine is required in the Introduction section, with definite examples of drawbacks of existing HRSV vaccines. Besides that, the authors have not mentioned some published successful research devoted to the design of peptide T-cell epitope based vaccine constructs directed against hepatitis C virus infection, e.g.   Fournillier A. et al.  (2006) "Primary and memory T cell responses induced by hepatitis C virus multiepitope  long peptides",  Vaccine, 24(16):3153-3164. doi: 10.1016/j.vaccine.2006.01.039; Huang X.J. et al. (2013) "Cellular immunogenicity of a multi-epitope peptide vaccine candidate based on hepatitis C virus NS5A, NS4B and core proteins in HHD-2 mice", J Virol Methods, 189(1):47-52. doi: 10.1016/j.jviromet.2013.01.003; Belyavtsev A.N. et al. (2021) "Synthesis and analysis of properties of an immunogenic fragment from NS4A polypeptide of hepatitis C virus", Russ. J. Bioorg. Chem. 47(3):713-718, doi: 10.1134/S1068162021030031. 

2.Lane 61, Introduction. The phrase "Peptide-based vaccines are synthesized in vitro by utilizing known immunogenic amino acids, generally from B-cell or T-cell epitopes..." contains an erroneous assertion: amino acids cannot be immunogenic or non-immunogenic. Immunogenicity can be attributed to peptides (=protein fragments), which are or are not B- or T-cell epitopes. The authors may mean that T-cell epitopes with certain HLA specificities have certain amino caid residues in certain key positions; characteristics that allow T-epitope predictions.

3. Materials and Methods section does not contain enough information about sample characteristics and experimental conditions. First, criteria of the past HRSV infection in PBMC  donors are not disclosed.  Second, the exact content of peptide vaccine samples are not disclosed as well, together with the adjuvant characteristics. It should be taken in mind that the adjuvant can influence the character of the immune response, e.g. transfer it from Th2 to Th1 and vice versa. So, the adjuvant should be disclosed. Third, please check the amounts of peptide mixture, PBS and adjuvant in a sample delivered to 1 animal in a 100-mkL volume: are they correct (lanes 141-145)? Fourth, please define the source of HEp-2 cells (tissue and company, where you have purchased the cells) used for virus neutralization assay. Fifth, please define if you have used human anti-F monoclonal antibodies so that you have required goat anti-human IgG (I suppose, Horseradish peroxidase-labelled?) for their detection (Lanes 172-173). Sources of antibody preparations should be disclosed as well. Sixth, it is unclear, whether all interleukins have beed determined in one experiment or in parallel experiments. Please, define this.

4. Regarding the list of peptides that constitute the F9 mixture, the peptide F-19 (RARRELPRF) is a T-epitope for CD8+ , i.e. cytotoxic T-lymphocytes, while the other peptides are Th-epitopes. There is no explanation, why this peptide has been included in the mixture of Th-epitopes.  Cytotoxic T-lymphocyte activation by peptide F-19 may contribute to IFN-gamma production while stimulating PBMC by F-9 peptide mixture. However, this fact is not explained in the Results section.   

5. Lane 312: MHC class should be defined, and in this context, MHC does not relate to T-cells, but to other cell types.     

Comments on the Quality of English Language

Please check and correct some typing or grammar errors, it can be done automatically. 

Round 2

Reviewer 1 Report

Comments and Suggestions for Authors

The authors have gladly addressed the comments and have significantly improved the manuscript. Now, the manuscript is acceptable for publication in its present form.